# SPEED: Selective Prediction for Early Exit DNNs

## Abstract

Inference latency and trustworthiness of Deep Neural Networks (DNNs) are the bottlenecks in deploying them in critical applications like autonomous driving. Early Exit (EE) DDNs overcome the latency issues by allowing samples to exit from intermediary layers if they attain 'high' confidence scores on the predicted class. However, the DNNs are known to exhibit overconfidence, which can lead to many samples exiting early and render EE strategies untrustworthy. We use Selective Prediction (SP) to overcome this issue by checking the 'hardness' of the samples rather than just relying on the confidence score alone. We propose SPEED, a novel approach that uses Deferral Classifiers (DCs) at each layer to check the hardness of samples before performing EEs. The DCs at each layer identify if a sample is hard and either differ its inference to the next layer or directly send it to an expert. Early detection of hard samples and using an expert for inference prevents the wastage of computational resources and improves trust. We also investigate the generalization capability of DCs trained on one domain when applied to other domains where target domain data is not readily available. We observe that EE aided with SP improves both accuracy and latency. Our method minimizes the risk by $50\%$ with a speedup of $2.05\times$ as compared to the final layer. The anonymized source code is available at https://anonymous.4open.science/r/SPEED-35DC/README.md

## 1 Introduction

The demand for Artificial Intelligence (AI) systems to automate decision-making is growing. However, their high memory, computational resource requirements, and the associated inference latency are bottlenecks in the deployment. Also, in socially sensitive or mission-critical machine learning applications, their trustworthiness is a concern (Kaur et al., 2022). For instance, the ability to know what you do not know and not get overconfident about it is essential. However, the issue of overconfidence in DNNs makes them vulnerable to wrong decisions and needs to be addressed.

To address the first issue, various adaptive inference methods have been developed, including Early Exit DNNs (EEDNNs) (Kaya et al., 2019; Zhou et al., 2020). EEDNNs use classifiers attached to the intermediary layers to perform adaptive inference, allowing samples attaining good confidence scores on the predicted class to exit at shallow layers with a label, thus reducing the average inference latency. To address the second issue, Selective Prediction (SP) (Chow, 1970) can be used, which allows a model to abstain from predictions when uncertain about its hardness. SP helps models better understand what they may not know, preventing them from being wrongly overconfident. We leverage SP to detect if a sample is hard for the shallow layers and decide either to defer the prediction to deeper layers or send it to an expert. This increases the trust in the model as incorrect decisions with high confidence are avoided.

The use of traditional SP methods Pugnana et al. (2024) for EEDNNs is challenging due to two major issues: 1) Existing methods use the confidence score output by the model to decide whether to abstain or not on each sample. However, the overconfidence issue hinders the purpose of the SP methods in effectively abstaining from the hard samples. This is illustrated in Figure 2a for the SST-2 dataset where we plot the average confidence of the samples in the true class across the layers using the trained EEDNN backbone. The experiment shows that multiple samples gain fake confidence

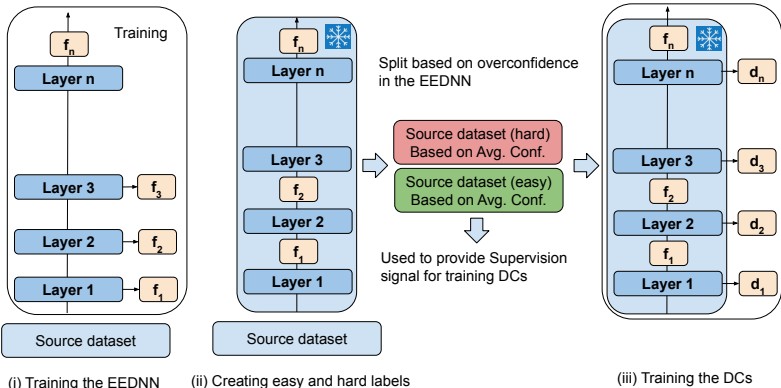

Figure 1: Flowchart of SPEED: i) The EEDNN backbone is trained with attached exit classifiers. ii) Split the dataset into easy and hard using the average confidence obtained from all the exit classifiers. iii) The splits are then used to train the DCs using labels as easy or hard.

even at the intermediary layers, and the confidence in the true class is very low. If existing methods are applied directly, they can get fooled by the model's fake confidence in these intermediary layers.

2) Existing methods make the decision to abstain after the entire model has processed the sample, which negates the speed benefits of EEDNNs. For example, in autonomous driving, if a decision to hand control over to a human driver is delayed until the entire model has been computed, critical time is lost. Thus, a more efficient approach is required to identify hard samples earlier in the process.

To overcome the issues, SP methods need to distinguish whether the model's confidence in a sample can be 'fake' or 'true' at the shallow layers and refer it to an expert instead of passing through higher layers. If a sample attains high fake confidence at the shallow layer, it is unlikely that the model's predictions would be better at deeper layers (see Appendix A.3). Thus, early referral of samples to experts saves crucial time and computation resources.

To address these issues, we propose a new framework named Selective Prediction for Early Exit DNNs (SPEED). This framework introduces deferral classifiers (DCs) at each layer of an EEDNN to detect if a sample is hard and the model can get overconfident, thus helping EEDNNs maintain the speed advantage while avoiding making inferences on samples that it does not understand. To train the DCs, we use samples that are labeled as 'easy' or 'hard' based on the average confidence they attain across the layer of EEDNN (see Subsection 3.4 for more details).

During inference, at each layer, the attached DC classifier detects if a sample is easy or hard. If hard, it exits the sample from DNN and send it to an expert. If found easy, it is sent to the EE classifier attached to the same layer. If the EE classifier completes the inference, the sample exits with a label, else it is sent to the next layer where the process repeats. Figure 1 shows the flowchart of SPEED.

Our method is also robust to the domain changes of the input samples. The reason for its better performance across diverse domains with similar tasks is the domain shifts do not change the semantic structure of the sentence. For instance, consider an IMDB review (source domain): "The movie would have been good if there was a better climax!," which has negative sentiment, but the model can say it is positive with high confidence due to the occurrence of the word 'good.' A similar review from an hotel review "The place would have been good if the food quality was better." also achieves fake confidence. Our framework, once trained on the source domain, can also handle similar fake confidence issues on the target domain. This reduces the need for another round of training for the same task but with a domain shift. In summary, our main contributions can are as follows:

- We introduce a new framework, SPEED, that uses Deferral Classifiers (DCs) at each layer of EEDNNs to make the inference fast and efficient.

- We develop a new strategy to train DCs. Each training sample is labeled as easy and hard utilizing its training dynamics across the layers of DNN.

- Our method generalizes well to various domains with minimal loss in performance making it robust to domain shifts of incoming samples during inference.

## 2 RELATED WORKS

**Early Exit (EE) techniques** allow models to make early predictions based on input complexity, have been widely applied across various domains, including image classification Teerapittayanon et al. (2016); Huang et al. (2017); Kaya et al. (2019); Wang et al. (2019b); Wołczyk et al. (2021), natural language processing (NLP) Xin et al. (2020); Zhu (2021); Zhou et al. (2020); Bajpai & Hanawal (2024), and image captioning Fei et al. (2022); Tang et al. (2023). These methods are known for their strong generalization capabilities, as demonstrated by models like CeeBERT (Bajpai & Hanawal, 2024). A key differentiator among existing approaches lies in the choice of confidence measures, such as prediction consistency Zhou et al. (2020), ensemble methods Sun et al. (2021), and output entropy Liu et al. (2021). Additionally, different training strategies, including separate training Xin et al. (2020) and joint training Zhou et al. (2020), have been explored to enhance the efficiency of EE methods. EE techniques have also been applied in practical settings like distributed inference Bajpai et al. (2024), where models are deployed across devices with varying computational capacities, such as mobile, edge, and cloud environments.

**Selective Prediction (SP)**, also referred to as selective classification or the reject option, has been extensively explored across various domains. Chow (1970) introduced a cost-based rejection model, analyzing the trade-off between errors and rejections. This concept has also been widely studied in the context of Support Vector Machines (SVMs) Brinkrolf & Hammer (2018); Hendrickx et al. (2024), nearest neighbors Hellman (1970), and boosting Cortes et al. (2016), demonstrating its versatility across different classification paradigms. In the domain of neural networks, LeCun et al. (1989) introduced a rejection strategy based on output logits, comparing the highest and second-highest activated logits to guide rejection decisions. Geifman & El-Yaniv (2017) proposed a selective classification technique to achieve a target risk with a specified confidence-rate function, laying the groundwork for risk-controlled predictions.

Recent advancements in SP have focused on architectural innovations Cortes et al. (2024) Geifman & El-Yaniv (2019). Other approaches, such as Deep Gamblers Liu et al. (2019) and Self-Adaptive Training Huang et al. (2020), incorporate an additional class for abstention. Feng et al. (2022) critically examined the selection mechanisms of these models and highlighted their limitations. Their findings suggest that the improved performance of these models is largely due to the optimization process leading to a more generalizable model, which in turn enhances SP performance.

Furthermore, several works have proposed deferral-based approaches, where the decision to predict or defer is determined by a cost function. These methods either assign prediction costs equal to model loss or defer predictions at a user-defined cost Okati et al. (2021); Mozannar et al. (2023); Verma et al. (2023).

Our method differs from existing approaches in three main ways: (1) To the best of our knowledge, we are the first to apply SP to EEDNNs. (2) We accelerate the deferral process by eliminating the need for every sample to pass through the entire backbone. (3) Our method of deferral decision is more accurate due to the integration of multiple DCs. (4) The separate training of these DCs does not affect the optimality of the backbone and enables better generalization across different domains, making our approach more robust to minor domain shifts during inference.

## 3 METHODOLOGY

We first motivate the issues of overconfidence in EEDNNs using the SST-2 dataset and discuss how Defferal Classifiers (DCs) can improve their inference time and their trustworthiness

### 3.1 MOTIVATION

In Figure 2a, we show the average of confidence score and its variance for SST-2 samples recorded across all the exits on the true class of the trained Early Exit BERT. The samples are classified into three categories based on their confidence scores: 1) Confident/Easy: The model is confident over the true class. 2) Confused: The model sees similar scores in both classes and is unsure about the class. 3) Fake: the model is overconfident (fake confidence) in the false class of these samples. Since most of the SP methods take this confidence score to decide to abstain, they end up accepting the wrong prediction from the model.

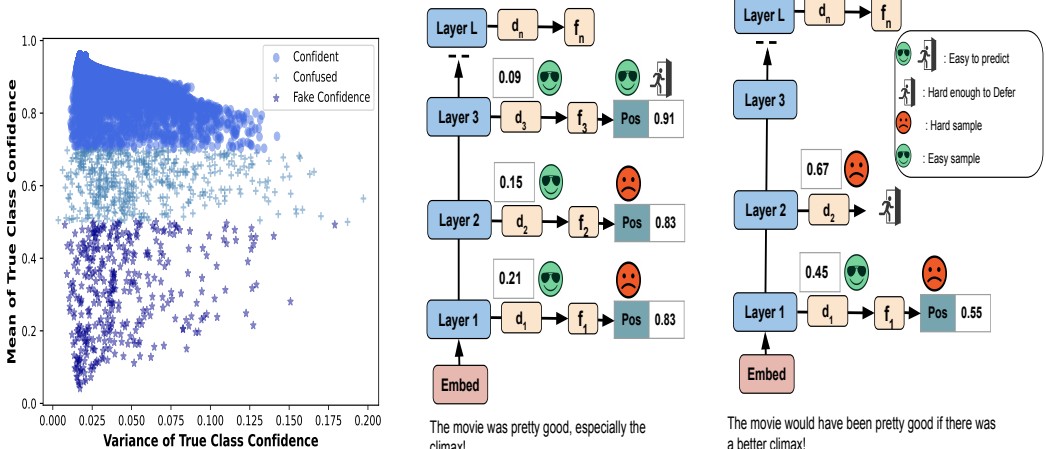

(a) Average confidence values on SST-2          (b) Examples of fake and true confidence.

Figure 2: The left figure shows the average confidence on the SST-2 dataset on the EEBERT back-bone. The right figure shows the effectiveness of our method during inference where an easy sample is exited early with a prediction a sample with fake confidence is detected and the model defers. The threshold for exiting is $0.90$ and the threshold for the DC score is $0.65$.

In our method, we explicitly train the DCs to catch such samples where the model has fake confi-dence. Note that the samples with higher variance are the ones where the model is highly confused and fluctuates between the true and false classes. A high DC score indicates that it is a hard sample, while a high score of the exit classifier indicates it has high confidence in a class.

In Figure 2b, we contrast easy and fake samples and how the use of DCs helps in handling overcon-fidence. The sample in the left figure is an easy sample. This is supported by low scores assigned by DCs and high scores by ECs. In the right one, the sample is hard. The ECs assign high confidence (fake), but the DCs give high scores, indicating that the confidence is fake. Thus with the use of DCs the risk of accepting the fake confidence during the selective prediction is significantly reduced. We add a discussion for such samples inA.3 and add more of these in Table 4.

The main challenge is training the DCs to identify the fake sample accurately. We use the Selective Prediction framework to build DCs for the EEDNNs.

## 3.2 SELECTIVE PREDICTION SETTING

We begin with the general SP setting. Let $\mathcal{X}$ be the feature space of the dataset $\mathcal{D}$ and $\mathcal{Y}$ be the label set. Let $P(\mathcal{X}, \mathcal{Y})$ be the distribution over $\mathcal{X} \times \mathcal{Y}$. A model $f : \mathcal{X} \to \mathcal{Y}$ is called a prediction function and its true risk w.r.t. $P$ is $R(f) := E_{P(X,Y)}[l(f(x), y)]$ where $l : \mathcal{Y} \times \mathcal{Y} \to \mathbb{R}^+$ is any given loss such as the 0/1 loss. Given the labeled dataset $\mathcal{D} = \{(x_i, y_i)\}_{i=1}^m \subseteq (\mathcal{X} \times \mathcal{Y})$ sampled i.i.d. from $P(X, Y)$, the empirical risk of classifier $f$ is $\hat{r} = \frac{1}{m} \sum_{i=1}^m l(f(x_i), y_i)$.

A selective model is a pair $(f, d)$ where $f$ is a prediction function and $d : \mathcal{X} \to \{0, 1\}$ is the deferral function that serves as the binary qualifier for $f$ as follows:

$$(f, d)(x) := \begin{cases} f(x) & if \quad d(x) = 1 \\ \text{defer} & if \quad d(x) = 0 \end{cases} \tag{1}$$

This equation shows the general setting of the selective models. The performance of a selective prediction model is quantified using coverage and risk. Coverage is defined as $\phi(g) := E_P[d(x)]$ and the selective risk of $(f, d)$ is defined as:

$$R(f, d) := \frac{E_P[l(f(x), y)(d(x))]}{\phi(d)} \tag{2}$$

Their empirical counterparts are given as $\hat{\phi}(d|\mathcal{D}) := \frac{1}{m} \sum_{i=1}^m d(x_i)$ and

$$\hat{r}(f, d|\mathcal{D}) := \frac{\sum_{i=1}^m l(f(x_i), y_i)(d(x_i))}{\hat{\phi}(d|\mathcal{D})}, \tag{3}$$

respectively. The goal is to find a pair $(f, d)$ such that $\hat{r}(f, d|\mathcal{D})$ is minimized.

We adapt this setting to our case with EEDNNs. We attached a Deferral Classifier (DC) in addition to the Early Exit classifier (EC) at each layer of DNN. Let $d_i$ and $f_i$ denote the DC and EC at the $i$th layers. Their predictions are based on the confidence scores described as follows. For $x \sim \mathcal{D}$, let $\mathcal{P}_i(c)$ denote class probability that $x$ belongs to class $c \in \mathcal{C}$ by the EC atlayer $i$. We define confidence score, denoted $C_i$, as the maximum over the class probability, i.e. $C_i := \max_{c \in \mathcal{C}} \hat{\mathcal{P}}_i(c)$. The decision to exit with a prediction at the $i$th layer is made based on the value of $C_i$. For a given threshold, $\alpha$ (exit threshold), if $C_i \geq \alpha$ the sample will be assigned a label $\hat{y} = \arg\max_{c \in \mathcal{C}}(\hat{\mathcal{P}}_i(c))$ and the sample is not further processed. $\alpha$ models the accuracy-efficiency trade-off.

For DCs, the confidence score is obtained as the output of the linear layer, which takes hidden representations of the intermediary layer as input. Let $S_i$ denote the confidence score at the $i$th DC. For a given threshold $\beta$ (deferral threshold), a sample is deferred at the $i$th layer if $S_i \geq \beta$. $\beta$ models the risk-coverage trade-off. A higher value of $\beta$ will have a lower risk with lower coverage and vice versa. We define the function $d_i(x)$ as:

$$d_i(x) := \left\{ \begin{array}{ll} 1 & if \quad S_i < \beta \\ 0 & otherwise \end{array} \right. \tag{4}$$

Hence at every layer the tasks of the pair $(f_i, d_i)$ could be written as:

$$(f_i, d_i)(x) := \left\{ \begin{array}{lll} \text{defer} & if & d_i(x) = 0 \\ f_i(x) & if & d_i(x) = 1, C_i \geq \alpha \\ i \leftarrow i+1 & if & d_i(x) = 1, C_i < \alpha \end{array} \right. \tag{5}$$

where $\alpha$ is the threshold that decides the early exiting of samples and $\beta$ named as the risk threshold is the hyperparameter that models the risk-coverage trade-off, the higher value of the risk factor increases the risk as well as coverage and vice versa.

When a sample is deferred it is exited from the DNN and assigned to an expert for prediction. The coverage for the pair $(f_i, d_i)$ could be defined as $\phi(d_i) := E_P[d_i(x)|C_i \geq \alpha]$ and the total coverage across all the layers could be defined as $\phi(G) = \sum_{i=1}^{n} \phi(d_i|C_i \geq \alpha)$ The selective risk of $(f_i, d_i)$ and the over all risk are defined as

$$R_i(f_i, d_i) := \frac{E_P[l(f_i(x), y)(d_i(x))|C_i \geq \alpha]}{\phi(d_i)} \quad \text{and} \quad R = \sum_{i=1}^{n} R_i(f_i, d_i). \tag{6}$$

Similar to 3, we can define the empirical selective risk, and our goal is to minimize it over the possible values of $(\alpha, \beta)$. Note that the total risks discourage sending all samples to an expert and the pair $(\alpha, \beta)$ together decide the overall risk-accuracy trade-off of the model.

Our framework starts with a pre-trained DNN and obtained an EEDNNs with DCs using the the following three steps. 1) Trains the ECs attached at each layers with frozen back-bone parameters. 2) Prepare the samples with appropriate labels to to train the DCs. 3) Trains the DCs attached to each layer with frozen back-bone and EC parameters. We next discuss each of these steps.

### 3.3 TRAINING EXIT CLASSIFIERS

Let $\mathcal{D}$ represent the data distribution with label space $\mathcal{C}$ used for training the backbone. The layers of the DNN are denoted as $L_1, L_2, \ldots, L_n$, with EC $f_i$ attached to layer $L_i$. For a given layer $L_i$, the hidden state $\mathbf{h}_i$ is computed as $\mathbf{h}_i = L_i(\mathbf{h}_{i-1})$, where $\mathbf{h}_0 = embedding(x)$. Each EC maps the hidden representations to class probabilities, i.e., $y_i = f_i(\mathbf{h}_i)$. The loss for the $i$th exit classifier could be written as:

$$\mathcal{L}_i = \mathcal{L}_{CE}(f_i(\mathbf{h}_i), y^*) \tag{7}$$

where $y^*$ is the true label and $\mathcal{L}_{CE}$ denotes the cross-entropy loss. We learn the parameters for all the ECs simultaneously following the approach outlined by Kaya et al. (2019), with the overall loss function defined as $\mathcal{L} = \frac{\sum_{i=1}^{n} i \cdot \mathcal{L}_i}{\sum_{i=1}^{n} i}$ where $n$ is the number of layers in the backbone. The weighted average considers the relative inference cost of each EC. Subsequently, the backbone parameters along with all the EC parameters.

## 3.4 TRAINING DATA FOR DCS

We perform supervised training of DCs using samples that are labeled as easy and hard. To obtain the labels, we leverage the confidence score of ECs attached to the backbone.

For a sample $x$ with label $y^*$, let $\mathcal{P}_i(y^*|x)$ denote the class probability $f_i$ assigns to $x$ on class $y^*$. The average true class probability score of the sample across all the layers is $\hat{\mu}_x = \frac{1}{n}\sum_{i=1}^{n} \mathcal{P}_i(y^*|x)$, and the variability is $\hat{\sigma}_x = \frac{\sum_{i=1}^{n}(\mathcal{P}_i(y^*|x_i)-\hat{\mu}_x)^2}{n}$. We sort the samples as per their scores $\hat{\mu}_x$ and label first-33% of the sorted samples as hard and the remaining as easy ones. We do not use the variance as a metric since high variance is only for the samples that fluctuate and might not include the samples where the model is highly confident on the wrong class. However, in the split, most of the high variance samples are included as hard samples as samples with high variance often exhibit low overall confidence.

The rationale behind creating these labels for DC training is rooted in the observation that hard samples across similar tasks share structural patterns, with semantic differences that are less relevant to the model. This allows our approach to generalize effectively across different domains.

## 3.5 TRAINING THE DCS

We attach the DCs at every layer of EEDNN whose parameters are frozen. The DCs can be any neural nets whose task is to map the hidden representations at every layer to a score of hardness in the range of 0 and 1 where a higher score means a harder sample. The loss for the $i$th DC is:

$$\mathcal{L}_i^{DC} = \mathcal{L}_{CE}(DC_i(h_i), z_i) \tag{8}$$

where $z_i$ is the binary label of the sample with 0 denoting and 1 denoting hard sample. We learn the parameters of all the DCs simultaneously. The overall loss function could be written as $\mathcal{L}^{DC} = \frac{\sum_{i=1}^{n} i \cdot \mathcal{L}_i^{DC}}{\sum_{i=1}^{n} i}$. The higher weight to deeper layers is based on the intuition that the deeper layer's hidden representations have high-level knowledge of the sample providing more critical information to the DC at that layer.

Once the DC training is complete, its parameters are frozen. Using the validation data, the parameters $(\alpha, \beta)$ are chosen that minimize the overall empirical risk defined in (6).

## 3.6 INFERENCE

For a test input $x \sim \tilde{\mathcal{D}}$ where $\tilde{\mathcal{D}} = \mathcal{D}$ when the source and target domain are the same, and for the given thresholds $\alpha$ and $\beta$, let the sample is processed till the $i$th layer. At the $i$th layer, its $S_i$ value is calculated using DC. If $S_i \geq \beta$, the sample is deferred and sent to an expert. If $S_i < \beta$, then the sample is referred to EC in the same layer, and its confidence score $C_i$ is checked. If $C_i \geq \alpha$, then the sample exits with a label completing the inference, else the sample is taken into the next layer, and the process continues till the sample reaches the last layer.

If the sample reaches the final layer, then the decision to infer or defer depends only on the confidence score of EC. If $C_i \geq \alpha$, then the sample is assigned a label. Otherwise, it defers the sample irrespective of the confidence of the DC. Hence, at every layer (except the final layer) of the backbone, every sample is either inferred, deferred, or passed on to the next layer.

# 4 EXPERIMENTS

In this section, we provide details of the experimental setup, key findings and analysis of our work.

## 4.1 DATASET

We utilized most of the GLUE Wang et al. (2019a) and the ELUE Liu et al. (2021) datasets. We evaluated SPEED on different datasets covering four types of classification tasks. The datasets used for evaluation are:

| Model | BERT | BERT-SR | SelNet | Cal-RF | PABEE-SR | Ours |
|---|---|---|---|---|---|---|
| Data | | | | Risk/Coverage | | |
| SST2 | 06.3/100 | 04.7/91.8 | 04.1/91.5 | 04.6/94.5 | 05.0/**95.6** | **03.5**/95.4 |
| IMDB | 10.4/100 | 05.2/86.4 | 04.8/85.3 | 04.1/85.1 | 05.5/86.7 | **03.9/87.2** |
| Yelp | 07.2/100 | 02.1/80.0 | 01.9/79.2 | 01.9/78.9 | 02.1/81.4 | **01.7/81.9** |
| SciTail | 09.7/100 | 03.3/84.5 | 03.5/84.0 | 03.1/83.8 | 03.7/85.2 | **03.1/85.9** |
| MRPC | 14.5/100 | 08.6/72.7 | 08.2/73.2 | 07.2/70.8 | 10.7/**75.3** | **06.5**/74.1 |
| QQP | 10.7/100 | 06.4/87.4 | 05.8/86.8 | 06.1/87.9 | 06.6/88.5 | **05.6/89.2** |
| MNLI | 15.5/100 | 09.3/86.3 | 08.9/85.2 | 08.8/84.7 | 09.7/87.0 | **08.3/87.2** |
| SNLI | 10.7/100 | 07.9/82.4 | 07.5/81.8 | 07.3/80.3 | 08.2/83.1 | **06.5/83.8** |
| Avg. Risk/Cov. | 10.6/100 | 06.3/84.2 | 05.9/83.4 | 05.8/83.7 | 06.8/85.7 | **05.2/86.1** |
| Avg.Speed | 1.00x | 1.00x | 1.00x | 1.00x | 1.71x | **2.05x** |

Table 1: In-Domain results: Results on the BERT backbone where the test and train set have the same distribution. We report the risk, coverage, average risk (Avg. Risk), average coverage (Avg. Cov.) and average speedup (Avg. Speed).

**1) Sentiment classification:** IMDB is a movie review classification dataset and Yelp consists of reviews from various domains such as hotels, restaurants etc. iii) SST-2 is also a similar type of dataset with the sentiment analysis task.

**2) Entailment classification:** We have used the SciTail dataset created from multiple questions from science exams an web sentences. MRPC (Microsoft Research Paraphrase Corpus) dataset which also has a semantic equivalence classification task of a sentence pair extracted from online news sources. We also perform experiment on the QQP (Quora Question Pair) dataset used to

**3) Natural Language Inference task:** We have used the MNLI and SNLI datasets for NLI tasks. SNLI is a collection of human-written English sentence pairs manually labeled for balanced classi- fication with labels *entailment, contradiction* and *neutral*.

In the Appendix 5, we also include the image datasets such as CIFAR-10 and Caltech-256 for im- age classification where CIFAR-10 has objects from 10 different categories while Caltech-256 has objects from 257 different categories.

## 4.2 EXPERIMENTAL SETUP

We have three parts in our experimental setup that are as follows:

**i) Training the EE backbone on the source dataset:** Initially, we train the backbone on the source dataset. We add a linear output layer after each intermediate layer of the BERT/RoBERTa model whose task is to map the hidden representation to class probabilities. We run the model for 5 epochs. We perform a grid search over batch size of $\{8, 16, 32\}$ and learning rates of $\{$1e-5, 2e-5, 3e-5, 4e-5 5e-5$\}$ with Adam Kingma & Ba (2014) optimizer. We apply an early stopping mechanism and select the model with the best performance on the development set. The experiments are conducted on NVIDIA RTX 2070 GPU with an average runtime of $\sim 3$ hours and a maximum run time of $\sim 10$ hours for the MNLI dataset.

**ii) Creating the dataset for DC training:** After training the EE backbone, we freeze the parameters of the backbone and then calculate the average confidence and variance of confidence across all the exit points. The dataset is then sorted in ascending order of confidence. The top-33% samples are provided the hard label and the remaining samples are treated as the easy ones. After creating the dataset, we attach the DCs i.e., a single linear layer mapping the hidden representations to hardness score. A hard sample will have a score closer to 1 and an easier sample will have a score close to 0.

The training for DCs is performed for additional 3 epochs. After training the DCs, the model can classify or defer early at each layer. For training the hyperparameters $\alpha$ we perform a grid search over the set $\{0.75, 0.8, 0.85, 0.9, 0.95\}$ and for $\beta$, we choose the set as $\{0.55, 0.6, 0.65, 0.7, 0.75\}$. The thresholds could be chosen based on the user requirements of risk and coverage. We set the thresholds as the ones that show similar trade-offs for comparison with existing baselines. We also plot the risk-coverage trade-off and discuss in the Appendix A.1 (see Figure 3a, 3b).

**iii) Inference:** We perform the inference on the test split of the source dataset and to show the generalization capabilities of our model, we also perform inference on a target domain dataset.

| Data/Model | BERT | BERT-SR | SelNet | Cal-RF | PABEE-SR | Ours |
|---|---|---|---|---|---|---|
| Src. - Tgt. | | | Risk/Cov. | | | |
| SST-IMDB | 19.8/100 | 14.9/86.0 | 14.2/85.1 | 14.6/87.0 | 14.8/86.4 | **12.8/87.2** |
| SST-Yelp | 17.9/100 | 11.0/78.3 | 11.4/79.0 | 11.9/80.4 | 13.7/81.7 | **09.0/82.0** |
| IMDB-SST | 18.1/100 | 15.5/85.2 | 14.8/86.0 | 15.1/85.5 | 16.2/86.2 | **13.8/85.9** |
| IMDB-Yelp | 24.3/100 | 15.2/81.5 | 15.0/79.8 | 14.6/80.8 | 15.5/80.1 | **13.5/82.1** |
| Yelp-IMDB | 22.8/100 | 13.2/73.4 | 12.6/72.7 | 12.9/72.7 | 14.8/73.9 | **09.5/74.5** |
| Yelp-SST | 25.1/100 | 13.8/71.5 | 14.0/72.1 | 13.6/70.6 | 14.2/73.8 | **12.0/75.3** |
| SNLI-MNLI | 25.1/100 | 14.9/80.7 | 14.5/79.2 | 13.9/78.5 | 15.2/81.6 | **12.3/82.8** |
| MNLI-SNLI | 19.8/100 | 11.4/76.8 | 10.8/74.8 | 11.3/76.2 | 12.6/**77.9** | **09.5**/77.4 |
| MRPC-SciTail | 35.7/100 | 25.6/74.2 | 24.1/71.6 | 25.9/**75.4** | 21.5/72.8 | **18.8**/73.5 |
| SciTail-MRPC | 29.9/100 | 23.3/69.9 | 22.7/67.8 | 23.1/69.2 | 24.7/70.3 | **21.4/70.9** |
| **Avg. Risk/Cov.** | 21.3/100 | 15.8/77.7 | 15.4/76.8 | 15.7/77.6 | 16.3/78.4 | **13.2/79.1** |
| **Avg.Speed** | 1.00x | 1.00x | 1.00x | 1.00x | 1.57x | **1.98x** |

Table 2: Out-Of-Domain results: Results on the BERT backbone where the test set has a different distribution from the training set (Src. (source)- Tgt. (target)). We report the risk, coverage, average risk (Avg. Risk), average coverage (Avg. Cov.) and average speedup (Avg. Speed).

Also, to maintain consistency with previous methods, we use the speed-up ratio as the metric to assess our model's improvement in speed as compared to existing methods. The speed-up ratio could be written as: $\frac{\sum_{i=1}^{n} n \times x_i}{\sum_{i=1}^{n} i \times x_i}$ where $x_i$ denotes the number of samples exiting from the $i$th layer and $n$ denotes the number of layers in the backbone. Note that the $x_i$ consists of the samples that either get an early inference or early deferral.

## 4.3 BASELINES

In this section, we detail the various baselines that we consider:

**1) BERT:** This baseline is where we perform vanilla BERT inference, where there is no selective prediction or early exiting. Hence the coverage in this case will always be full.

**2) BERT-SR:** It uses softmax-response where at the final output of the BERT model, the softmax layer is added after the final layer of the BERT and if the model is confident enough on the prediction, then it infers the sample, else it abstains.

**3) SelectiveNet:** In this there is an additional loss component to lower the risk of the model, however the criteria to abstain is similar to the BERT-SR method.

| Data/Mdl | RBTa | RBTa-SR | PBEE-SR | Ours |
|---|---|---|---|---|
| | | Risk/Coverage | | |
| SST2 | 7.8/100 | 6.2/92.7 | 6.8/93.8 | **4.5/94.1** |
| Yelp | 5.9/100 | 5.1/93.2 | 5.4/**94.0** | **4.2**/93.9 |
| IMDB | 10.1/100 | 7.5/91.5 | 7.1/92.1 | **6.8/92.6** |
| SciTail | 8.5/100 | 7.0/**90.2** | 6.8/89.4 | **5.7**/89.7 |
| MRPC | 13.3/100 | 9.6/87.5 | 10.1/88.6 | **9.0/88.6** |
| QQP | 9.8/100 | 8.4/91.9 | 8.1/93.2 | **7.5/93.7** |
| **Avg.** | 9.23/100 | 7.3/91.6 | 7.3/91.8 | **6.2/92.1** |
| **Avg. Spd.** | 1.00x | 1.00x | 1.73x | **2.19x** |

Table 3: Results over the RoBERTa backbone.

**4) Calibrator model:** This baseline considers a calibrator i.e., the output of the model is passed to a different smaller model, in our case, we use a random forest to decide about abstaining the sample.

**5) PABEE SR:** We attach the softmax-response to the PABEE model, where the exits are attached, and there is a softmax layer at the final layer used to decide to abstain or perform inference.

All the parameters are kept the same as given in the codebases of the respective baselines. For a fair comparison, the risk-coverage trade-off hyperparameter was chosen based on the values that minimize the risk over the validation dataset for all the baselines.

## 4.4 EXPERIMENTAL RESULTS

In Table 1, Table 3, we present the results of the test split of the training set on the BERT and RoBERTa backbone respectively. Each experiment is performed five times and the average results are reported (we put the stability of our results in Appendix 6). We can observe that our method outperforms the existing baselines. the justification for lower risk is due to the fact that our method

| IMDB | | | |
|---|---|---|---|
| **Example** | **True lbl.** | **Fake Avg. Conf.** | **DC** |
| I was really excited about this film, but I was wrong. | negative | 0.96 | 0.87 |
| His last film was better than this. | negative | 0.89 | 0.82 |
| Low budget, but creepy enough to hold your interest | positive | 0.91 | 0.84 |
| Don't judge it by bad cover picture. | positive | 0.88 | 0.89 |
| SST-2 | | | |
| mysterious and brutal nature | positive | 0.90 | 0.91 |
| the under-7 crowd | negative | 0.89 | 0.79 |
| seems endless | negative | 0.85 | 0.86 |
| a movie with two stars | positive | 0.83 | 0.85 |

Table 4: Examples of samples achieving fake confidence, the table shows an example, its true label (True lbl.), the confidence of the model on the wrong class (Fake Avg. Conf.) and the deferral classifier score that abstains the sample early.

catches the samples that have fake confidence early and abstains from predicting them while existing methods predict such samples. A few examples of such samples are given in Table 4 where the model starts gaining fake confidence but the DCs identify them and force them to defer (more discussion in Appendix A.3). There is also significant speedup in our method, which comes from the early exiting of easier samples and early detection of the hard samples. The combined speedup is $2.05x$ the vanilla BERT inference saves crucial time and significantly reduces the computational cost. The coverage of PABEE-SR is sometimes higher than our method and other baselines, the reason being the early classifiers that infer the sample and it is not processed till the final layer. However, it has a higher risk as compared to the existing methods.

In Table 2, we show the results of the different domains i.e., the model is tested on a different domain. Here we can show the generalization capabilities of our method as compared to the existing ones. Observe that when the domain of the test dataset changes, our method has the least performance drop i.e., the least increase in speedup. The improvement in risk values of our method is more significant when the domain changes as compared to the existing methods. This comes as the effect of training the DCs using the easy and hard samples that help them generalize well to other domains with similar tasks.

BERT-SR, PABEE-SR and Cal-RF use the confidence values available at the final layer to decide to abstain from a sample. Since the confidence could be fake, the risk of these models goes high. The lower performance of the SelectiveNet model is due to the fact that it optimizes the abstaining process using the backbone parameter which affects the optimality of the backbone.

Note that the better coverage of our method is due to the reduction of the overthinking issue. Overthinking is caused when the model loses confidence in an easy sample when it is processed deeper into the backbone due to the extraction of overly complex features Kaya et al. (2019); Zhou et al. (2020). In such cases, our method early infers the sample improving the coverage and since existing only checks the confidence at the final layer that has dropped due to overthinking and end up abstaining from predicting the sample. This improves the performance as well as coverage of our method pushing multiple metrics simultaneously.

## 5 CONCLUSION

In this work, we introduced SPEED, a selective prediction tailored to improve inference latency and overconfidence issues in EEDNNs. We introduce new classifiers named deferral classifiers (DCs) to identify if a sample is hard for a given layer and if the model can get overconfident about its false class. Thus, it helps the model to understand what it does not know, which is crucial to prevent the model from making a false prediction with high confidence! We develop a method to train the DCs to effectively identify if a sample is hard or easy to infer at a given layer.

Our experiments demonstrate that SPEED consistently outperforms existing baselines in both risk reduction and speedup. Further, SPEED generalizes well across different domains, minimizing the need for retraining and making it robust to domain shifts.

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

# A  APPENDIX

## A.1  RISK-COVERAGE TRADE-OFF

In Figure 3a and 3b, we show the risk-coverage trade-off, where the values are obtained by changing the risk hyperparameter in our case and for the existing baselines, we use the hyperparameters used in their codebases to obtain the results. Figure 3a shows the in-domain trade-off where the model is trained in the SST2 dataset and tested on the test split of the same dataset while in Figure 3b we plot the trade-off when the model is trained on the SST2 dataset and tested on the IMDB. We can observe that our method has a significantly smaller drop in coverage, and the risk decreases. The drop in coverage is lower when the domain of the dataset changes for our method. Note that better performance of our method comes as multiple DCs help to effectively catch the samples where the model can gain fake confidence or is unable to gain confidence. Also, mitigation of overthinking issues through early exits helps improve coverage pushing multiple metrics simultaneously.

## A.2  RESULTS FOR THE IMAGE TASKS

In table 5, we show the results of our model on the Cifar10 and Caltech-256 datasets on the MobileNet model. We attach exit classifiers in a similar fashion. In this setup, we first train the model with early exits and then the DCs are trained. After this step, the clean test set images are used for inference (shown as pristine). Then we add noise to the images of the test split and consider that as a domain change, where the noise added is the Gaussian blur. The level of Blur depends on the value of $\sigma$, higher the value of $\sigma$, the more blur the image is. This translates to the scenarios of domain change in autonomous driving where the environment factor might change the distribution of incoming samples.

The performance of our method is similar in the image domain as well where our method outperforms all the existing baselines with a significant margin. This further proves the effectiveness of our method.

## A.3  SOME EXAMPLES OF FAKE CONFIDENCE

In table 4, we list some examples where the model gains fake confidence i.e., it is confident over the wrong class. For instance, consider the example *a movie with two stars* with confidence across 12 layers as $[0.47, 0.21, 0.03, 0.01, 0.03, 0.06, 0.42, 0.26, 0.22, 0.13, 0.09, 0.11]$, this sample is even confusing for humans as it has two meanings one is a movie that consists of two-star actors or it means a movie which can be rated with only two stars out of five, due to which the model gains fake confidence over the wrong class. Note that the DCs catch the hard samples as the threshold for DCs is kept smaller, and at every layer before checking the confidence of the prediction, we check the score of hardness given by the DC at that layer.

Samples such as *Don't judge it by bad cover picture* give the model an overall negative impact that makes it overconfident towards the negative class as the model focuses on some keywords which could be *bad* in this case making a negative impact on the model. The confidence over the true class is $[0.47, 0.20, 0.08, 0.05, 0.10, 0.07, 0.09, 0.01, 0.03, 0.02, 0.01, 0.008]$.

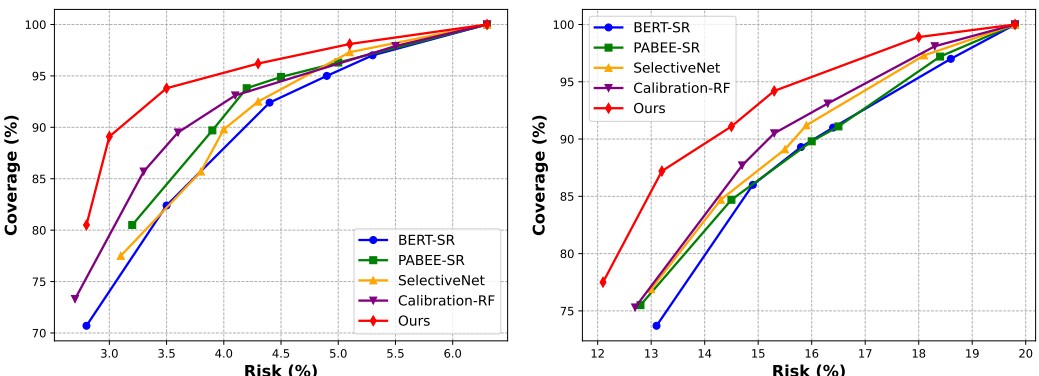

(a) Risk-coverage trade-off for SST2-SST2.    (b) Risk-coverage trade-off for SST2-IMDB.

| Data/Model | MNet | MNet-SR | SelNet | PABEE-SR | Ours |
|---|---|---|---|---|---|
| | | Caltech-256 Risk/Coverage | | | |
| Pristine | 21.7/100 | 16.5/89.8 | 15.3/87.7 | 15.9/88.5 | **14.8/89.3** |
| $\sigma = 0.5$ | 24.8/100 | 17.2/**86.0** | 16.8/85.4 | 16.3/85.1 | **15.9**/85.9 |
| $\sigma = 0.75$ | 26.2/100 | 19.8/84.6 | 18.5/82.6 | 19.0/84.9 | **17.1/85.1** |
| $\sigma = 1.0$ | 28.4/100 | 21.2/82.3 | 20.9/**81.9** | 20.4/81.3 | **18.5**/81.8 |
| | | Cifar10 - Risk/Coverage | | | |
| Pristine | 7.5/100 | 4.8/93.4 | 3.9/92.2 | 4.9/**95.8** | **4.1**/95.3 |
| $\sigma = 0.5$ | 10.1/100 | 8.3/91.9 | 6.2/89.6 | 7.6/92.6 | **5.9/93.5** |
| $\sigma = 0.75$ | 14.9/100 | 10.7/89.0 | 9.4/88.1 | 9.8/88.9 | **8.5/89.4** |
| $\sigma = 1.0$ | 21.7/100 | 16.2/83.9 | 14.5/82.4 | 15.2/84.8 | **13.8/85.2** |

Table 5: Results on CIFAR10 and Caltech-256 datasets with different levels of noise on the MobileNet backbone.

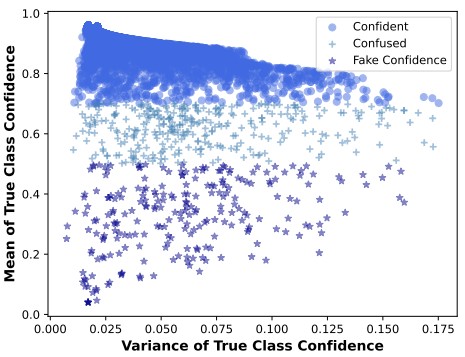
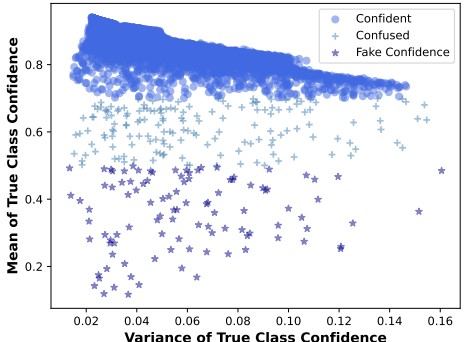

(a) Average confidence values on IMDB dataset.    (b) Average confidence values on SciTail dataset.

Figure 4: Average confidence values on SciTail and IMDB datasets on the EEBERT backbone

| | BERT-SR | SelNet | Cal-RF | PABEE-SR | Ours |
|---|---|---|---|---|---|
| | | | Risk/Coverage | | |
| SST2 | 4.7±0.07 | 4.1±0.03 | 4.6±0.02 | 5.0±0.06 | 3.2±0.04 |
| IMDB | 8.2±0.09 | 7.8±0.05 | 8.1±0.06 | 9.5±0.10 | 6.9±0.03 |
| Yelp | 2.1±0.04 | 1.9±0.02 | 1.9±0.03 | 2.1±0.03 | 1.7±0.02 |
| SciTail | 3.3±0.06 | 3.5±0.03 | 3.1±0.05 | 3.7±0.04 | 3.1±0.02 |
| MRPC | 8.6±0.10 | 8.2±0.05 | 7.2±0.09 | 10.7±0.09 | 6.5±0.03 |

Table 6: Stability of our method as compared to others.

