# OpenReview forum: "SPEED: Selective Prediction for Early Exit DNNs"
_ICLR.cc/2025/Conference — ICLR 2025 Conference Withdrawn Submission_

### Official Review · Reviewer_ZZzM · 2024-10-23

**Soundness:** 2
**Presentation:** 2
**Contribution:** 3
**Rating:** 3
**Confidence:** 3

**Summary:**

The authors propose a new framework named Selective Prediction for Early Exit DNNs (SPEED) which tackles the topics of inference latency and trustworthiness of DNNs. On the one hand, early exit DNNs overcome the latency issues by allowing samples to exit
from intermediary layers if they attain ‘high’ confidence scores on the predicted class. On the other hand, selective prediction is used to overcome the issue of overconfidence, by checking the ‘hardness’ of the samples rather than just relying on the confidence score alone. To this end, they use deferral classifiers at each network layer to check the hardness of samples before performing early exits. Moreover, the authors investigate the generalization capability of their method.

**Strengths:**

- The authors explore an exciting theme whereby runtime/cost is reduced during inference while considering prediction confidence.
- I like the idea of referring certain samples to an expert instead of passing it through higher layers when the probability that the network will still predict the sample correctly is too low.
- I like that the method can also be applied to domain shifts and works comparatively well.
- Large selection of tasks and datasets.

**Weaknesses:**

- I find it confusing that examples from autonomous driving are mentioned all the time, but the focus is more on language tasks and classification.

The paper is not written very clearly:
-  The first description of figure 2a is unclear to me. It sounds like you also obtain information about different layers instead of a complete average value as shown.
- The SST-2 dataset is already used in the first and third sections, but not even briefly explained for which task the dataset is and what it looks like.
- Until the fourth section, it is unclear to me which problems the presented method is/can be applied to.
- There is no reference to appendix A.2 and figure 3 has no caption or naming.

There are some mistakes in the paper, here are a few examples:
- "Defferal Classifiers" in line 152
- "Similar to 3," in line 250, eq. 3
- "back-bone" and "backbone"
- "DCs using the the" in line 253

**Questions:**

- It is often mentioned that certain samples are submitted to an expert, what does the expert look like?
- A short explanation of Early Exit BERT is missing on first use. It is not a baseline so was Early Exit BERT trained by the authors for the first time or is it a different name for their own method?
- "The rationale behind creating these labels for DC training is rooted in the observation that hard samples across similar tasks share structural patterns, with semantic differences that are less relevant to the model." - How did you find that?
- Only the PABEE-SR baseline has an early exit, were there no other baselines?

---

### Official Review · Reviewer_cg4J · 2024-10-29

**Soundness:** 3
**Presentation:** 2
**Contribution:** 3
**Rating:** 3
**Confidence:** 4

**Summary:**

This paper proposes SPEED (Selective Prediction for Early Exit DNNs), a novel framework that combines selective prediction with early exit DNNs to address both inference latency and trustworthiness issues. The key innovation is introducing Deferral Classifiers (DCs) at each layer to detect hard samples early and either defer them to an expert or pass them to deeper layers. Experimental results show SPEED achieves 50% risk reduction with 2.05× speedup compared to final layer inference, while also demonstrating good generalization across different domains.

**Strengths:**

- The paper addresses a significant problem in DNN deployment, where both inference latency and model trustworthiness are crucial challenges that need to be solved.
- The proposed SPEED framework achieves impressive results in handling both latency and trustworthiness issues, demonstrating 50% risk reduction and 2.05× speedup compared to baseline methods.
- The method shows strong generalization capability across different domains and tasks, with comprehensive validation on both NLP and computer vision applications.

**Weaknesses:**

The paper is difficult to follow due to multiple writing issues, for example:

## 1. Illogical Flow Between Paragraphs

### A. Issues in Introduction
The paper states:
```
"To address the first issue, various adaptive inference methods have been developed, including Early Exit DNNs (EEDNNs)... To address the second issue, Selective Prediction (SP) can be used...The use of traditional SP methods for EEDNNs is challenging due to two major issues:..."
```
Key problems:
- Jumps to "first issue" and "second issue" without clear definition
  * The paper should first clearly state what these two issues are
  * A proper introduction would begin with "DNN deployment faces two major challenges: latency and trustworthiness"

- No explanation for why EE and SP need to be combined before discussing challenges
  * The paper should explain why using either solution alone is insufficient
  * Need to demonstrate why combining these methods would be beneficial


### B. Issues in Section 3.1 (Motivation)
The subsection begins:
```
"In Figure 2a, we show the average of confidence score and its variance for SST-2 samples recorded across all the exits on the true class of the trained Early Exit BERT."
```
Key problems:
- Lacks clear problem introduction
  * Should first explain why confidence analysis is important

- Abruptly transitions to their method
  * Should first explain what patterns or issues were observed in the analysis
  * Then explain why these observations motivate their approach


## 2. Technical Presentation Issues
- Figure 1: No explanation of symbols fn and dn in figure or caption
  * These are key components of the method and should be clearly defined
  * Caption should include "fn represents Exit Classifier, dn represents Deferral Classifier"

- Line 152: Missing period

- Figure 2: Distorted aspect ratio and compressed fonts
  * Affects visual presentation and readability
  * Should maintain proper aspect ratio and font sizes for clarity

**Questions:**

## 1. Clarity and Organization

Q1. The introduction jumps directly to solutions without proper problem setup. Could you:
- Clearly define the two main issues (latency and trustworthiness) first?
- Explain why combining EE and SP is necessary?
- Add a paragraph discussing limitations of existing methods separately?

Q2. Section 3.1 (Motivation) starts abruptly with Figure 2a analysis. Could you:
- Explain why confidence score analysis is important?
- How do these observations directly motivate your method design?

## 2. Technical Details

Q3. Regarding the training process:
- Why choose 33% as the threshold for hard samples?
- How sensitive is the method to this threshold?
- Were other thresholds tested?

Q4. About model architecture:
- Could you clarify the roles of fn and dn in Figure 1?
- How are the features shared between Exit Classifier and Deferral Classifier?
- Why use this specific architecture design?






Your responses to these questions would help readers better understand the technical contributions of your work.

---

### Official Review · Reviewer_KPCZ · 2024-10-31

**Soundness:** 2
**Presentation:** 3
**Contribution:** 2
**Rating:** 5
**Confidence:** 4

**Summary:**

This paper introduces SPEED (Selective Prediction for Early Exit DNNs), a framework that addresses both inference latency and trustworthiness issues in Deep Neural Networks with early exits. The key innovation is the introduction of Deferral Classifiers (DCs) at intermediate layers, which assess whether a sample is "hard" before allowing early exit decisions. Unlike traditional approaches that rely solely on confidence scores, SPEED identifies potentially misleading high-confidence predictions and either defers them to deeper layers or routes them to an expert. The framework includes a new training strategy for DCs using samples labeled as easy or hard based on their behavior across network layers. Through extensive experiments on text and image classification tasks, SPEED demonstrates superior performance over existing baselines, achieving 50% risk reduction with 2.05× speedup compared to full inference. Notably, the approach generalizes well across different domains without requiring retraining, making it particularly robust to domain shifts in real-world applications.

**Strengths:**

The paper presents a novel combination of selective prediction and early exit DNNs that addresses the critical challenges of inference latency and overconfidence in deep neural networks. The introduction of deferral classifiers at intermediate layers represents an original approach to detecting hard samples early, rather than relying on traditional confidence scores alone.
The authors propose a comprehensive set of experiments across multiple datasets and domains. The evaluation covers both in-domain and cross-domain scenarios, with metrics showing consistent improvements in risk reduction and speedup compared to baselines.
The paper presents well-structured methodology explanations and effective use of visualizations. The flowcharts and examples effectively illustrate key concepts like fake confidence and the impact of deferral classifiers.
The authors provide a thorough review of related work in both early exit techniques and selective prediction approaches. The related work section effectively traces the development of these ideas from foundational papers to recent advances.
The proposed work has broad significance for practical AI deployment, addressing key challenges in latency and trustworthiness. The promise to generalize across domains without retraining would make the approach particularly valuable for real-world applications. Results shown on both text and image tasks, suggest AI systems could be made more robust and reliable.

**Weaknesses:**

The paper has several areas that need improvement to strengthen its contribution. The choice of 33% threshold for labeling samples as hard in Section 3.4 appears arbitrary and lacks theoretical justification. The authors should explore how different thresholds affect performance and provide guidance for selecting this parameter in practice.

The treatment of expert deferral needs more detail. While the paper frequently mentions deferring to experts, it doesn't address practical considerations like expert availability, cost, or what happens when experts disagree. This is particularly important for the autonomous driving use case mentioned in the introduction.

The computational overhead of adding deferral classifiers at each layer is not thoroughly analyzed. While the paper shows improved inference speed, it should quantify the additional training time and memory requirements of the DCs compared to baseline approaches.

Overall, the weaknesses discussed so far are not showstoppers. The main concern is the code provided which upon examination does not provide confidence it could have generated the results shown in the paper. The code itself does not run, and with the example given in the README file throws an error:

    handle = open(
FileNotFoundError: [Errno 2] No such file or directory: 'data/SST2/SST2.csv'

Examining the code, it is not obvious where the described procedure took place: "We add a linear output layer after each intermediate layer of the BERT/RoBERTa model whose task is to map the hidden representation to class probabilities. We run the model for 5 epochs.
We perform a grid search over batch size of {8, 16, 32} and learning rates of {1e-5, 2e-5, 3e-5, 4e-
5 5e-5} with Adam Kingma & Ba (2014) optimizer."

**Questions:**

1. In section 3.4, the paper mentions labeling the top-33% samples as hard based on confidence scores. However, in the code's 'evaluate' function, the hardness labeling appears to use both mean confidence and variance thresholds (alpha and beta). Could you clarify how exactly the 33% threshold mentioned in the paper relates to these parameters?

2. For the deferal classifier training, the code shows a BCE loss being used, but the paper doesn't specify the loss function. What was the rationale for choosing BCE loss over other options for training the deferal classifiers?

3. The paper reports a speedup of 2.05x compared to vanilla BERT, but the code calculates speedup using '12*len(data_loader.dataset)/sum(exit_lis)'. Could you explain if this matches how the speedup was calculated for the paper results?

4. In Table 1, the paper shows coverage percentages in the 70-100% range, but looking at the code's evaluate_disc function, the coverage calculation appears different from how it's described in the paper. Could you clarify how exactly the coverage metrics in Table 1 were computed?

5. The paper mentions that each experiment was performed five times and reports average results, but the code doesn't seem to have explicit support for multiple runs. How were the multiple runs managed and results aggregated?

6. The model appears to have a parameter 'num_exits' set to 12, but the paper doesn't explicitly state this architecture detail. Were all reported results obtained using exactly 12 exit points?

These clarifications would help ensure reproducibility of the paper's results.

---

### Official Review · Reviewer_ttnz · 2024-11-02

**Soundness:** 2
**Presentation:** 2
**Contribution:** 2
**Rating:** 3
**Confidence:** 4

**Summary:**

The paper introduces SPEED, a novel framework designed to enhance the efficiency and reliability of Early Exit Deep Neural Networks (EEDNNs) by addressing two main issues: overconfidence in shallow layers and inference latency. SPEED integrates Deferral Classifiers (DCs) at each layer to detect “hard” samples that require further processing, ensuring early exits only for samples the model classifies with confidence. By identifying and deferring complex samples early, authors claim that SPEED improves both speed and accuracy, ultimately reducing computational demands and increasing trust in model predictions.

**Strengths:**

- The writing and presentation are good and easy to follow.
- The reported improvement in inference time is promising.
- Extensive evaluation across diverse datasets and settings both in-domain and out-of-domain.

**Weaknesses:**

- Training and maintaining DCs at every layer would add substantial overhead in both training and inference time. For deep models, such as those with 50 or more layers, the requirement to train and deploy a DC for each layer could be unfeasible, especially in resource-constrained environments. Additionally, the presence of DCs in every layer could increase latency during inference, as each DC would introduce an additional computational step, especially if the model needs to check exit points layer by layer. This would counteract the goal of accelerating inference in early exit DNNs, as the added latency from the DCs might outweigh the benefits of early exiting.

=> Action: Instead of placing DCs at every layer, the authors can strategically place DCs at a subset of layers, such as those where significant feature extraction occurs (e.g., after each major block in a ResNet). This approach would reduce the number of DCs, lowering computational demands while still enabling early exits. Alternatively, they could implement a shared classifier that functions across multiple layers, potentially using attention mechanisms to adjust its predictions based on the features of each layer. This way, a single DC could dynamically adapt to different layers, reducing the need for individual classifiers at each exit point.

- Each additional classifier would add to the memory and storage requirements, particularly if each DC is stored and loaded independently. This could pose significant challenges for deploying SPEED on edge devices or in environments with limited storage capacity.

=> Action: Authors can consider consolidating DCs by sharing model parameters across similar layers or using lightweight DC architectures to reduce memory consumption. Additionally, they could explore compression techniques, such as quantization or pruning, to make DCs more storage-efficient for deployment on resource-constrained devices. Each additional classifier increases memory and storage demands, particularly if DCs are stored and loaded independently, which can pose challenges for edge devices or environments with limited storage capacity.

- The paper does not fully explore how the architecture of DCs impacts performance, such as whether simpler models could maintain accuracy while further reducing latency.

=> Action: Include an ablation study on DC architectures (e.g., using simpler classifiers or different configurations) to determine the impact of model complexity on efficiency and accuracy.

**Questions:**

Check weakness.

---

### Official Review · Reviewer_eo3W · 2024-11-03

**Soundness:** 2
**Presentation:** 2
**Contribution:** 2
**Rating:** 5
**Confidence:** 3

**Summary:**

This paper proposes Deferral Classifiers at each layer of EEDNNs to improve inference efficiency. The authors suggest training these classifiers based on the "hardness" of samples, defined by the true class confidence score from the trained exit classifier at each layer.

**Strengths:**

- The proposed technique is reasonable, as the deferral classifiers are trained based on information of the true class confidence score instead of the max confidence score, which could lead to overconfidence.
- The writing is generally clear.

**Weaknesses:**

- The baseline methods only consider last-layer output; early-exit methods as baselines would better support the efficiency claims for the proposed method.
- The claim at line 80, "If a sample attains high fake confidence at the shallow layer, it is unlikely that the model’s predictions would be better at deeper layers," is debatable. High "fake confidence" refers to confidence in a wrong class prediction, and it is possible that early layers, which may focus on low-level features rather than semantic features, yield poor results on tasks that require deeper-layer information. Thus, overconfidence at lower layers does not necessarily indicate poor predictions at deeper layers.
- Training the deferral classifier based on the true class confidence is similar to the approach in [1]; discussing this similarity and highlighting any differences would strengthen the paper.
- An important ablation study on the impact of removing the deferral classifiers is missing.
- References for the baseline methods are missings.
- Several typos are present:
  - Misspelled method names in Table 3
  - Line 220: "atlayer i"
  - Line 244: missing period

[1] Corbière, Charles, et al. "Addressing failure prediction by learning model confidence." Advances in Neural Information Processing Systems 32 (2019).

**Questions:**

1. What is the performance when using only the exit classifier (EC) without the deferral classifier (DC), or only the DC without the EC? A more detailed ablation study is necessary.
2. How does the proposed method perform compared to methods that consider early-exit options rather than only the last-layer output? The current baselines only consider last-layer output, which makes for an unfair comparison, especially when evaluating efficiency.
3. As the method still requires validation data, what if the EC classifiers are calibrated with the validation data to mitigate the overconfidence issue, can the authors compare to this to see if the proposed deferral classifier is still more effective or not?

---

### Note · Authors · 2024-11-27

I have read and agree with the venue's withdrawal policy on behalf of myself and my co-authors.